# Ovarian Neuroglial Choristoma in a Bitch

**DOI:** 10.3390/vetsci9080402

**Published:** 2022-08-01

**Authors:** Eleonora Brambilla, Barbara Banco, Stefano Faverzani, Paola Scarpa, Alessandro Pecile, Debora Groppetti, Claudio Pigoli, Marco Giraldi, Valeria Grieco

**Affiliations:** 1Department of Veterinary Medicine and Animal Science, University of Milan, Via dell’ Università 6, 26900 Lodi, Italy; eleonora.brambilla@unimi.it (E.B.); stefano.faverzani@unimi.it (S.F.); paola.scarpa@unimi.it (P.S.); alessandro.pecile@unimi.it (A.P.); debora.groppetti@unimi.it (D.G.); 2Mylav-Laboratorio Analisi La Vallonea, 20017 Passirana di Rho, Italy; barbarabanco@laboratoriolavallonea.net (B.B.); m.giraldi.vet@gmail.com (M.G.); 3Laboratorio di Istologica, Sede Territoriale di Milano, Dipartimento Area Territoriale Lombardia, Istituto Zooprofilattico Sperimentale della Lombardia e dell’Emilia Romagna, Via Giovanni Celoria 12, 20133 Milano, Italy; claudio.pigoli@izsler.it

**Keywords:** dog, choristoma, neuroglial choristoma, ovary, ovarian choristoma

## Abstract

**Simple Summary:**

Choristomas are rare malformations consisting of normal mature tissue in an abnormal location. Neuroglial choristomas consist of heterotopic mature neural tissue and, in human medicine, they are predominantly reported in the head and in the neck, except for one recent case reported in a foot of a child. In domestic animals, neuroglial choristomas are exceedingly rare, reported only in the retina of a dog, in the pharynx and in the skin of two kittens, and within the oropharynx of a harbor seal. In this report we describe the first ovarian case of neuroglial choristoma in canine species. The lesion was present in a three-year-old intact female Jack Russell Terrier that, seven years after ovariectomy, is still clinically normal. Together with a recent case described in the foot of a child, this case confirms that neuroglial choristoma may also be found far from the skull or spine, supporting the hypothesis that they may arise from an early embryological migration defect.

**Abstract:**

Neuroglial choristomas are rare malformations consisting of heterotopic mature neural tissue at a site isolated from the brain or spinal cord. In human medicine, neuroglial choristomas are predominantly reported in the head and in the neck, except for one recent case reported in a foot of a child. In domestic animals, neuroglial choristomas are exceedingly rare, reported only in the retina of a dog, in the pharynx and in the skin of two kittens, and within the oropharynx of a harbor seal. A three-year-old intact female Jack Russell Terrier presented for elective ovariectomy exhibited a cystic lesion 2 cm in diameter expanding in the right ovary. Histological examination of the lesion revealed a mass composed of well-organized neuroglial tissue. Immunohistochemistry with primary antibodies against GFAP, NSE, and IBA-1 confirmed the neuroglial origin of the mass. At the time of this writing, 7 years after ovariectomy, the dog was clinically normal. Together with a recent case described in the foot of a child, this case confirms that neuroglial choristoma may also be found far from the skull or spine, supporting the hypothesis that they may arise from an early embryological migration defect.

## 1. Introduction

Mature brain tissue at a site isolated from the cranium or spinal cord is termed neuroglial choristoma [1]. Decade after decade, this entity was improperly defined with various terms such as: “heterotopic brain tissue”, “glioma”, or “teratoma” [2,3,4,5], reflecting the controversy that exists regarding its pathogenesis [6,7,8]. In human medicine, neuroglial choristomas are considered malformations, with heterotopic neural tissue predominantly reported in the head and the neck [9]. In domestic animals, neuroglial choristomas are sporadic lesions reported only in the retina of a dog [10] and in the pharynx and the skin of two kittens [8,11]. In wildlife species, a perioral and oropharyngeal presentation was described in a neonatal harbor seal [7]. This report aims to describe histologically and immunohistochemically a case of ovarian neuroglial choristoma in a bitch.

## 2. Case Description

A three-year-old intact female Jack Russell Terrier was presented for ovariectomy to the Reproduction Unit of the Veterinary Teaching Hospital (Università degli Studi di Milano). The dog was regularly vaccinated, treated for both internal and external parasites, and fed with mono-protein commercial dry food for food-responsive enteropathy. Except for this enteric chronic disorder, diagnosed at one year of age and resulting in occasional episodes of vomit and inappetence, the bitch was deemed healthy based on physical examination and pre-anesthesiologic laboratory tests. Furthermore, no alteration of the reproductive cycle was reported.

Elective ovariectomy was performed at the anestrous phase using a standard technique through median laparotomy access with the bitch supine [12]. The removed ovaries were immediately sent to the Pathology Unit for histological examination.

Grossly, the left ovary was normal, showing multiple yellowish corpora lutea in regression, whereas the right one was expanded by a 2 cm-diameter cystic lesion filled with clear fluid and lined by a solid gray tissue rim. Both ovaries were fixed in 10% buffered formalin for three days and then trimmed and routinely processed for histology. Briefly, samples were dehydrated through graded alcohols, clarified in xylene, and embedded in paraffin wax. From paraffin blocks, 5 μm-thick sections were obtained and stained with hematoxylin and eosin (HE).

Histologically, the right ovary was expanded by a cystic lesion that was 2 cm in diameter, well-demarcated, not encapsulated, and consisting of well-organized neuroglial tissue characterized by multifocal degeneration and necrosis which partially replaced the ovarian cortex. Neuroglial tissue surrounded an optically empty cystic cavity that was internally lined by ependymal tissue organized to form a choroid plexus. The lesion was composed by fibrillary eosinophilic collagen bundles in which glial cells were present. Among the glial cells, three cell types were detectable. One type was represented by spindle cells with indistinct cell borders, small amounts of pale eosinophilic cytoplasm, round nuclei, vesicular chromatin, and inconspicuous nucleoli resembling astrocytes. The second cell type consisted of spindle to polygonal cells with a perinuclear halo and densely basophilic round nuclei resembling oligodendrocytes. The third cell type consisted of spindle cells with densely basophilic, elongated or cigar-shaped nuclei resembling microglial cells. Within glial cells, other stellate, large cells with distinct cell borders, abundant basophilic cytoplasm, and a central round nucleus with coarsely-stippled chromatin and one prominent nucleolus, namely, neuronal bodies, were visible. The elongated portion of the neurons (axons) variably wrapped in myelin sheath (myelinated fibers) were also detectable. Multifocally within the neuroglial tissue there were extensive coalescing areas of rarefaction and cavitation with the loss of neuroparenchyma and small focal areas of liquefactive necrosis with the replacement of the neuropil by granular eosinophilic necrotic debris, foamy reactive gitter cells multifocally containing intracytoplasmic granular red-brownish material (ceroid), fibrin, hemorrhages, and edema. Multifocally increased vacuolization defined as spongiosis, dilated myelin sheaths containing swollen and degenerate axons forming spheroids, dilated myelin sheaths containing necrotic debris and gitter cells (ellipsoids and digestion chambers), and slight gliosis were also visible. Scattered reactive, variably stellate cytoplasmic projections and multinucleated giant astrocytes with nuclei located at the periphery of the cell, often polarized to one side of the cytoplasm and defined as gemistocytic astrocytes, were also visible. Neurons were multifocally characterized by central chromatolysis (with peripheral displacement of the Nissl substance), by peripheral chromatolysis (with the presence of the Nissl substance only around the nucleus and absence at the periphery), or by neuronal swelling or shrinkage (necrosis and degeneration). Occasional necrotic neurons with total loss of nucleus and the Nissl substance were also present. Scattered neurons showed small, optically empty peripheral vacuoles in their cytoplasm defined as vacuolar degeneration. Finally, the cystic cavity was internally lined by a single layer of ciliated polygonal to cubical cells resembling ependymal cells, often arranged in papillary structures projecting into the lumen and supported by lose connective tissue and capillaries, such as choroid plexus (Figure 1).

To confirm the neuroglial origin of the cystic lesion, serial sections were further obtained and examined immunohistochemically by the avidin–biotin–peroxidase complex method (ABC; Vector Laboratories, Burlingame, CA, USA) technique [13] with primary antibodies listed in Table 1. Serial sections of canine brain were used as positive controls, whereas for negative controls, to confirm the specificity of the markers, replicate sections were incubated with isotype-specific immunoglobulins (mouse IgG for NSE and rabbit IgG for both GFAP and Iba-1, both reagents from Vector Laboratories, Burlingame, CA, USA) (Table 1) [14].

Tissue samples demonstrated a good immunohistochemical reactivity with the expected marker expression in positive controls. Glial fibrillary acidic protein (GFAP) and ionized calcium-binding adapter molecule 1 (IBA-1) labelling highlighted the astrocyte-type cell and microglial cell cytoplasm and cytoplasmatic branching processes, respectively (Figure 1b,d), whereas the neuron-specific enolase (NSE) strongly labelled the cytoplasm of neuron-type cells (Figure 1c).

According to the histological appearance and immunohistochemical results, the cystic lesion was diagnosed as neuroglial choristoma.

### OUTCOME

Abdominal ultrasound examination performed about a month after ovariectomy revealed only a mild hyperechogenicity at the left and right ovarian sites. These findings were considered within the limits of normality after surgery. No abdominal visceral nor parietal lymphadenomegaly was detected. No lesions were detected in the liver, spleen, kidneys, adrenal glands, urinary bladder, or uterus peritoneum.

No therapies were prescribed after histological diagnosis. Three years after ovariectomy, the dog underwent enterotomy to remove a foreign body ingested during an episodic pica secondary to the worsening of enteropathy. In that circumstance, no record referable to ovarian neuroglial choristoma emerged. Therefore, ovariectomy was considered curative.

At the time of this writing, 7 years after ovariectomy, the dog is clinically normal based on communications with the owner. 

## 3. Discussion

In human as in veterinary medicine, ovarian choristoma is extremely rare, except for ectopic adrenal glands which have been widely documented in humans, horses, and cats [15,16,17,18]. The presence of areas of adrenal tissue in the gonads is justified by the common embryologic origin of the cortical adrenal gland and gonadal ridge [19]. In human medicine, neuroglial choristomas are exceedingly rare findings of heterotopic neural tissue reported predominantly in the head and neck, in the middle-ear structures, and recently, in the foot of a child [9,20,21].

Regarding the pathogenesis of neuroglial choristoma, many theories have been proposed in human medicine. The origin has been attributed to the herniation of a portion of fetal cerebral tissue, which can occur during embryonic development as in the development of encephaloceles with subsequent separation from the cranial cavity [5,22,23]. However, the failure to find the site of herniation led authors to suggest that the origin of glial choristomas was from neural crest cells in the head and neck, which can undergo neuroglial development [24]. Concerning glial choristomas of the tongue, the most reliable origin seems to be from the displaced neural tissue present early in occipital somites from which the tongue muscles originate. It seems that the nests of pluripotential cells become separated before the complete fusion of the neural tube and are brought to the extracranial tissues in association with normally migrating cells [25]. Alternatively, an error in early embryonic development may result in displaced neuroectodermal cells, which undergo differentiation and ectopic proliferation [9,26].

The presence of normal neuroglial tissue within the ovary of a bitch was previously described twice: in 2012 by Rota et al., and by Pires et al., 2019. In both these cases, the entity was diagnosed as ovarian mature monophasic teratoma [27,28]. However, the term teratoma is used in human and veterinary medicine to describe a benign tumor of the ovary composed of mature tissue deriving from at least two embryonic layers (ectoderm, mesoderm, or endoderm) [29]. Moreover, the term monophasic teratoma was used in the past in human ovarian pathology for malignant tumors derived from astrocytes, collectively called neuroectodermal tumors of the ovary [30]. With this stated, for the present canine case, the definition of neuroglial choristoma was considered appropriate as it was consistent with normal neuroglial tissue in an ectopic location. Together with a recent case described in the foot of a child [9], this case confirms that neuroglial choristomas may also be found far from the skull or spine, supporting the hypothesis that they may arise from an early embryological migration defect.

## 4. Conclusions

Neuroglial choristomas are exceedingly rare in domestic animals and they have been reported only in the retina of a dog and in the pharynx and in the skin of two kittens. In the present report we described, based on histological and immunohistochemical findings, the first case of ovarian neuroglial choristoma in canine species. This case confirms that neuroglial choristoma may also be found far from the skull or spine and suggests considering this lesion among canine ovarian pathologies.

## Figures and Tables

**Figure 1 vetsci-09-00402-f001:**
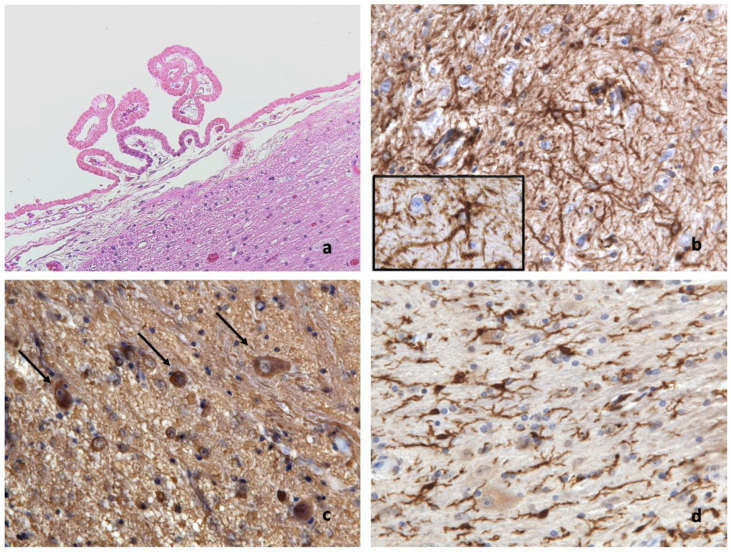
(**a**) Well-organized neuroglial tissue facing an optically empty cystic cavity internally lined by ependymal tissue organized to form a choroid plexus; H&E; 200×. (**b**) GFAP strongly and diffusely labels the fibrillary background and glial cell cytoplasm and branches (inset); IHC; 400×, inset 1000×. (**c**) NSE strongly labels the neuronal cytoplasm (arrows) and neuropil; IHC; 400×. (**d**) IBA-1 strongly labels microglial cell cytoplasm and cytoplasmatic branching processes; IHC; 400×.

**Table 1 vetsci-09-00402-t001:** Immunohistochemical (Ihc) examination details.

Ihc Marker	Antigen Retrieval	Primary Antibody	Positive Control	Code	Species
GFAP	None	Polyclonal, dilution: 1:3000; Dako, Carpinteria, USA	Internal: peripheral nerves	Z334	Rabbit
NSE	None	Monoclonal, dilution: 1:1000; Dako Carpinteria, USA	Internal: Cerebral cortex	IS612	Mouse
IBA-1	HIER; Buffer H pH 9	Polyclonal, dilution: 1:2000; Wako Corporation, USA	Internal: Cerebral cortex	019-19741	Rabbit

GFAP: glial fibrillary acidic protein; NSE: neuron-specific enolase; IBA-1: ionized calcium-binding adapter molecule 1.

## Data Availability

Not applicable.

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
