# Peer review of "Ovarian Neuroglial Choristoma in a Bitch"

_vetsci, 2022, doi:10.3390/vetsci9080402_

Round 1

Reviewer 1 Report

This paper is a case report and not a research article but the authors presented a very rare case of ovarian neuroglial choristomas that hasn't been reported in the dog, increasing this way its originality. 

The whole manuscript is well written, and easy to be followed by the reader. The text and structure of the manuscript are clear and require no changes from my point of view. 

Macroscopic findings were further investigated with the use of histological examination and immunohistochemistry, that confirmed the neuroglial origin of the mass and lead to the conclusion that neuroglial choristomas may also be found far away from the skull or spine if they arise from early embryological migration.

Author Response

We thank the reviewer for the good comments received.

Reviewer 2 Report

This paper deals with a quite rare case of a neoplasia of the ovary, which was clearly demonstrated to be neuroglial origin on immunohistochemistry. This information should be valueable for veterinary surgeons in the future.

The choristoma is a quite rare disease in dogs ( as well as human), thus worth to be reported. This paper just reported the histopathological analysis for the neoplasm identified on the ovary after ovariectomy in a bitch. 

It is valuable that this tumor lesion was identified on the ovary of dogs and the third case of choristoma in dogs.

Histopathological analysis is reasonable and almost demonstrated to be choristoma.

The paper was well written in proper English (though by my decision) and all the contents are easily understood.

Author Response

(The authors gave the same response as above.)

Reviewer 3 Report

The manuscript describes the original case of a neuroglial choristoma in the ovary of a bitch. The case is clear, well presented, the histologic description is well detailed and immunohistochemical characterization of the lesion is appropriate.

Some minor suggestions follow:

-       Line 56-60 : it is a too long sentence, which is difficult to read and understand. Please, amend it.

-       Line 60: could you describe if there is an arrangement among the different described cells?

-       Line 75 “dilated myelin sheath”: it is weird to read about Wallerian degeneration, while no mention was done about the presence of axons before. Maybe it would be clearer to describe before all the “normal“ structures and then all the degenerative changes.

-       “ghost neuron” is not very clear to me : do you mean empty spaces? Do you mean necrotic neurons?

-       Line 148 “within and ovary”: I think that “and” should be replaced with “the”

Figures need to be improved.

-       Figure 1 may benefit of a higher magnification;

-       Figure 2 and 3 are out of focus.

-       Figure 2 has no inset (despite what stated in the legend)

-       Maybe figure 2 and 4 are inversed, please verify

-       What is reported in the legend as a specific background, for me it is a noise background.

Author Response

Ref.3

The manuscript describes the original case of a neuroglial choristoma in the ovary of a bitch. The case is clear, well presented, the histologic description is well detailed and immunohistochemical characterization of the lesion is appropriate.

We thank the reviewer for the good comments received and for the suggestions as well. All suggestions have been followed (see below) and the related changes in the text has been highlighted in yellow (see below). Images were also improved

Some minor suggestions follow:

          Line 56-60: it is a too long sentence, which is difficult to read and understand. Please, amend it. – The sentence has been amended and divided in two sentences

Line 60: could you describe if there is an arrangement among the different described cells? We didn’t observe a particular arrangement among the different glial cells.

Line 75: “dilated myelin sheath”: it is weird to read about Wallerian degeneration, while no mention was done about the presence of axons before. Maybe it would be clearer to describe before all the “normal“structures and then all the degenerative changes. Thank you. Added according to your suggestion. Line 83-85

“ghost neuron” is not very clear to me: do you mean empty spaces? Do you mean necrotic neurons? Yes, and the text has been changed accordingly.

Line 148 “within and ovary”: I think that “and” should be replaced with “the” – corrected accordingly

Figure 1 may benefit of a higher magnification - The image was taken at higher magnification. We also apologize because the magnification of the previous image was not present in the first version of the manuscript

Maybe figure 2 and 4 are inversed, please verify – We apologize for the inversion. Fig.2 was effectively fig 4 in the first version of the manuscript (IBA1 IHC) –. Now the images have been inserted at the correct place and one of them also retaken (ex-fig2, now Fig. 4), in order to achieve a better focus as you suggested

Figures 2 and 3 are out of focus. Also the image number 3 was retaken to achieve a better focus.

Figure 2 has no inset (despite what stated in the legend) – the inversion of Fig. 2 and Fig. 4 caused the problem. The old fig 4, that now has correctly been named Fig.2, has the inset that was in the legend (up-left).

What is reported in the legend as a specific background, for me it is a noise background. – The legend of Fig. 3 (NSE IHC) has been amended focusing the positive labelling of neurons and neuropil. What seems a background is the positive neuropil, so we apologize for the previous unprecise legend.

Reviewer 4 Report

OVERVIEW AND SUMMARY

In this case report, the authors describe a rare malformation with an unusual location. The manuscript is well organized, clear and detailed, the literature is appropriate and updated, and the figures are appropriate.

Considering the rarity of the lesion, a thorough description and characterization are of interest. I list a few minor points.

MINOR COMMENTS

Figure 1: I see in the legend a few mistakes: in Fig. 1.2 is indicated an inset that I cannot see in the picture, while in Fig. 1.4, where there is an inset, it is not indicated in the legend. Could you please correct this part?

Moreover, it would be useful to insert an hpf inset of figure 1.2, to better appreciate the pattern of immunolabeling.

If it is possible, I also suggest adding a picture of the gross appearance of the lesion.

line 59: did you mean “empty”, instead?

Please, to improve transparency and reproducibility, add the codes of antibodies as well as the species they were produced in and the details of isotype-controls used.

Author Response

Ref. 4

In this case report, the authors describe a rare malformation with an unusual location. The manuscript is well organized, clear and detailed, the literature is appropriate and updated, and the figures are appropriate.

Considering the rarity of the lesion, a thorough description and characterization are of interest. I list a few minor points.

We thank the reviewer for the positive general comments and for the minor corrections suggested. All of them have been done (see below) and highlighted in green in the text.  

       MINOR COMMENTS

Figure 1: I see in the legend a few mistakes: in Fig. 1.2 is indicated an inset that I cannot see in the picture, while in Fig. 1.4, where there is an inset, it is not indicated in the legend. Could you please correct this part? We thank for the comment and apologize. Fig 2 and Fig 4 were inverted. The inset was correctly present in fig 2, that was erroneously put at the place of Fig. 4. In the second version of the manuscript figures have now the correct position.

Moreover, it would be useful to insert an hpf inset of figure 1.2, to better appreciate the pattern of immunolabeling. We thank for the suggestions, being now the fig 2 the ex-fig 4, it has the inset requested.

If it is possible, I also suggest adding a picture of the gross appearance of the lesion. We thank for the suggestion, but unfortunately, we have not a gross picture. We remember very well the gross appearance of the lesion: when the ovary was trimmed the lesion seemed a cyst with a wall partially thicker and a little bit gray wall. Seeming a cyst we unfortunately didn’t take any image!! It is a real pity!

Line 59: did you mean “empty”, instead? Yes and we corrected accordingly

Please, to improve transparency and reproducibility, add the codes of antibodies as well as the species they were produced in and the details of isotype-controls used: Added accordingly.
